# WFDroneBench: A Benchmark for Sensor Placement and Drone Routing for Wildfire Detection

## Abstract

Increasingly frequent and severe wildfires threaten ecosystems, public health, and infrastructure. Early detection is vital but limited by existing monitoring systems. Drones offer mobile, real-time coverage, but optimizing sensor placement and drone routing in dynamic fire zones remains challenging. To address this, we introduce WFDroneBench, an open-source Python benchmarking library for early wildfire detection that integrates machine-learned risk maps with optimization-based deployment strategies for sensors, charging stations, and drones. It evaluates risk maps, optimization strategies, and monitoring equipment using standardized metrics and realistic wildfire simulations. The framework supports benchmarking across predictive and decision-making components: machine learning researchers can assess risk models and operations research experts can compare routing strategies. WFDroneBench includes 7746 scenarios across 49 locations, built from historical ignitions, real-world wildfire risk maps, and simulated fire spread, along with two ground detector and four drone routing strategies. Our experiments show that risk-aware strategies – Team Orienteering Problem (TOP) and Max-Coverage – significantly outperform other baselines when risk maps are sufficiently accurate, with TOP achieving the fastest detection on the most difficult fires. We further find that risk-aware static infrastructure helps even under an imperfect riskmap and drone-based detection outperforms ground sensors. Finally, our results reveal two key open challenges: (i) detecting small fires rapidly and reliably, and (ii) improving risk-map prediction, where the gap between ground-truth ignition patterns and available risk maps highlights a significant opportunity for ML innovation.

## 1 Introduction

Wildfires and the resulting deforestation have been on the rise in recent years (7; 32; 18), causing substantial economic damage and costs exceeding \$70 billion annually (31). The 2018 California wildfires alone resulted in \$149 billion in total damages, including \$32 billion in health-related costs from mortality, medical care, and lost productivity (33). Despite increased wildfire awareness and resource investments, the January 2025 Los Angeles wildfires devastated the wildland–urban interface, causing \$150 billion in damages and evacuating 200,000 residents (12). This highlights critical gaps in current detection and response systems. Effective monitoring and early wildfire detection is therefore crucial to ensure fires remain manageable and to minimize damage.

Traditional approaches rely on ground sensors and satellites, but both have notable limitations. Ground sensors provide limited visibility and are costly to build and maintain, while satellites often lack the spatial and temporal resolution needed for timely detection (4). Notably, more than half of U.S. counties, particularly in rural regions, lack any air quality monitoring infrastructure (27). Drones are increasingly recognized as a promising solution for wildfire detection. Their ability to scan large, remote areas with high spatial and temporal resolution makes them well-suited for early fire detection (4; 14). Advances in onboard sensing and deep learning have significantly improved the accuracy of aerial wildfire detection, enabling drones to reliably identify smoke or flames (14; 20; 21; 38; 39). However, most studies in this domain assume that drones follow pre-defined flight paths. The critical question of how to actively plan drone routes to maximize early discovery of fires remains largely unaddressed.

Most existing work on drone routing focuses on post-detection monitoring and response in which the fire has already been located and the objective is to track its perimeter, coordinate suppression efforts, or support firefighter operations (see e.g., (15; 2; 16)). These reactive strategies assume prior knowledge of fire locations. In contrast, proactive wildfire detection, where drones search for unknown fires across large landscapes, is far less explored. Existing detection efforts largely center on the placement of ground sensor networks (3; 1; 5; 29) or on ML-driven image analysis to enhance onboard fire recognition by drones (see e.g., (26; 4)), neglecting the integration of active routing strategies. Yet, successful early detection demands tight coupling between intelligent routing and accurate sensing. This includes integrating mobile drones and stationary sensors, dynamically adapting to changing risk conditions, and operating under constraints like battery life, transmission range, and real-time data availability. Key challenges include: how to route drones to maximize early detection probability, how to adapt to evolving risk, and how to coordinate drone and sensor deployments under limited resources.

In this paper, we address the underexplored challenge of proactive wildfire detection. We introduce an integrated framework for optimizing and evaluating risk-aware sensor placement and drone routing, supported by a modular open-source Python library with standardized evaluation metrics and semi-synthetic benchmarking datasets. Our main contributions are:

1. **Introducing a problem formalism for wildfire detection with ground sensors and aerial drones.** We develop a mathematically-grounded framework for proactive wildfire surveillance that integrates evolving risk information, and real-world constraints such as drone battery life, recharging, transmission range, and multi-drone coordination. This enables designing adaptive, risk-aware routing strategies designed to improve the timeliness and effectiveness of wildfire detection efforts.

2. **`WFDroneBench`, a benchmarking framework for wildfire detection strategies.** `WFDroneBench` is a flexible, open-source framework that integrates wildfire risk data, drone profiles, and operational constraints. It supports the development and evaluation of detection strategies, risk maps, and drone specifications using diverse metrics, enabling controlled ablations and cross-method comparisons. The framework facilitates experiment reproducibility, visual diagnostics, and community-driven development, and supports benchmarking of both predictive and decision-making components across varied use cases.

3. **A realistically diverse, semi-synthetic wildfire benchmark dataset.** We curate a semi-synthetic dataset combining historical ignition records with physics-based wildfire spread simulations to represent fires spreading without human intervention. Scenarios span diverse geographic and ecological conditions, capturing variations in climate and weather conditions, terrain, ignition, wildfire, size and rate of propagation, all captured in rich metadata enabling detailed subgroup analysis for robust benchmarking.

4. **Developing, implementing, and benchmarking diverse wildfire detection strategies in a uniform code base.** We adapt, implement and evaluate eight wildfire monitoring strategies: two for ground sensor placement and four for drone routing. These include both heuristic and optimization-based methods, all implemented within a consistent Julia codebase. Our experiments reveal that high-accuracy risk maps are essential for enabling effective risk-aware strategies, risk-aware static infrastructure helps even under an imperfect riskmap, and drone-based detection outperforms ground sensors.

## 2 Related Work

Previous work on optimal drone routing for wildfire detection spans a range of modeling approaches. One line of work proposes mixed-integer linear programs (MILPs) to jointly optimize the placement of watchtowers or balloons alongside battery-constrained, single-route drone missions (9). While these methods note that drones can be reused, recharging is not explicitly modeled. The probabilistic path planning (PPP) strategy proposed in (34) uses logistic regression to estimate fire probabilities and subsequently employs a dynamic programming (DP) algorithm to generate an optimal routing for unmanned aerial vehicles (UAV). However, it does not explicitly consider operational constraints such as battery life, charging requirements, or return-to-base logistics. Other studies minimize the makespan, defined as the longest mission duration across all drones, motivated by constraints such as connectivity synchronization, coordinated data retrieval, and bandwidth limits; in some cases, this

is coupled with the placement of recharging stations (28; 19). These approaches are well-suited for structured surveillance or exhaustive coverage, where all regions are considered equally important. An alternative strategy, the risk-specific UAV patrol path (RSUPP), uses Gaussian mixture clustering to partition high-risk regions into subareas, assigning each UAV to the shortest route through one such region based on static wildfire risk (35). Other modeling approaches include nature-inspired metaheuristics such as Particle Swarm Optimization (PSO) (37). There are many approaches for generic UAV routing (see e.g. (24; 25; 10)), including the Traveling Salesman Problem (TSP) and its extension, the TOP. TSP emphasizes exhaustive coverage with minimal travel cost, while TOP focuses on selecting a subset of valuable locations to maximize collected reward. We build on the strategies applicable to our setup, using wildfire risk as reward.

We implement four proactive drone routing approaches, that leverage wildfire risk data, accounting for spatial heterogeneity and enabling adaptive replanning as conditions evolve. Unlike models assuming uniform importance or known fire locations, the incorporated methods adapt to evolving wildfire risk. As TOP and PSO are applicable to our framework, one of our drone routing approaches is based on a modified TOP strategy solved with a PSO algorithm. It serves as a first extensible framework supporting integration and comparison of a variety of routing and detection strategies.

## 3 PROBLEM FORMULATION

Our objective is to effectively deploy a limited number of ground sensors, charging stations, and drones across a real-world geographic region for early wildfire detection. To formalize the problem, we consider a spatiotemporal environment represented as a two-dimensional grid $\mathcal{I}$ of size $N \times M$ over a discrete time horizon $\mathcal{T} = \{1, \dots, T\}$. At each time step $t \in \mathcal{T}$, a fire may ignite and propagate to adjacent grid points. If a fire ignites at grid point $i$ and time $t_0$, it is considered detected once it enters the sensing radius of any deployed device at time $t_{\text{det}}$, resulting in a detection delay $\Delta t = t_{\text{det}} - t_0$.

To minimise the expected detection delay $\mathbb{E}[\Delta t]$ over the ignition distribution, we define a **detection strategy** that optimizes the placement of ground sensors and charging stations, denoted by $\{x_i^g, x_i^c\}$, and the routing of and charging of drones over time, denoted by $\{a_{its}, c_{its}\}$. This strategy is guided by a spatiotemporal wildfire risk map, which informs both infrastructure deployment and drone movement decisions. The fire risk is encoded as a burn risk field $r_{it} \in [0, 1]$, representing the probability that grid point $i$ is burning at time $t$, incorporating both ignition and spread probability. This risk map can be obtained from physical simulation models or predicted using machine learning methods that capture the underlying dynamics of wildfire behavior.

Devices may only be deployed at feasible subsets: ground sensors on $\mathcal{I}_g \subseteq \mathcal{I}$ and charging stations on $\mathcal{I}_c \subseteq \mathcal{I}$. We define binary placement variables $x_i^g \in \{0, 1\}$ and $x_i^c \in \{0, 1\}$ to indicate whether a sensor or station is placed at location $i$. A drone fleet $\mathcal{S}$ is routed through the domain according to binary variables $a_{its}$ and $c_{its}$, where $a_{its} = 1$ if drone $s$ flies at location $i$ at time $t$ and $c_{its} = 1$ if drone $s$ is charging at grid point $i$ at time $t$ (see notation overview in Table 1.) Device allocation is constrained by resource limits. Spatial exclusion constraints further prevent deployments from being placed too close to one another. Drones follow a single-hop kinematic rule—after spatial and temporal rescaling using the drone cruising speed, they may move only to neighboring cells at each time step—and must remain within a communication radius $R$ of a charging station. Each drone's endurance is limited to $D_{\text{max}}$ time steps and is fully replenished upon landing at a charging station.

## 4 WFDroneBench

**Design goal.** To bridge data-driven risk modeling with operational decision-making, we introduce WFDroneBench, an open-source benchmark for early wildfire detection. The framework supports the integration of machine learning–generated risk maps with optimization-based deployment strategies for ground sensors, charging stations, and drones. It includes tools for loading wildfire risk maps, simulating dynamic wildfire scenarios, implementing deployment strategies, and evaluating early detection performance through standardized metrics. WFDroneBench is modular and extensible. Users can implement custom deployment policies or integrate their own risk maps and fire scenarios to evaluate end-to-end performance. The framework supports benchmarking across both predictive and decision-making components across diverse use cases: ML researchers can benchmark risk

| Sets | Description | | Variables | Description |
|---|---|---|---|---|
| $\mathcal{I}$ | Set of all possible grid points on the map | | $x_i^g \in \{0,1\}$ | 1 if a ground sensor is placed at grid point $i$ |
| $\mathcal{I}_g$ | Set of all feasible grid points for the ground sensors | | $x_i^c \in \{0,1\}$ | 1 if a charging station is placed at grid point $i$ |
| $\mathcal{I}_c$ | Set of all feasible grid points for the charging stations | | $a_{its} \in \{0,1\}$ | 1 if drone $s$ flies at grid point $i$ at time $t$ |
| $\mathcal{I}_d$ | Set of all feasible grid points for the drones | | $c_{its} \in \{0,1\}$ | 1 if drone $s$ is charging at grid point $i$ at time $t$ |
| $\mathcal{T}$ | Total time horizon with cardinality $T$ | | $\theta_{it} \in \{0,1\}$ | 1 if grid point $i$ is covered by a drone at time $t$ |
| $\mathcal{H}$ | Optimization time horizon with cardinality $H$ | | $b_{ts} \in \mathbb{Z}$ | Battery of time $s$ at time $t$ (unit time) |
| $\mathcal{S}$ | Set of drones | | | |

| Parameters | Description |
|---|---|
| $r_{it}^d$ | Dynamic wildfire risk in grid point $i$ at time $t$ |
| $r_i^s$ | Static wildfire risk in grid point $i$ |
| $T_{\text{reev}}$ | Reevaluation step |

Table 1: Notation used in the problem formalism and the formulation in Section 5.1.

models via their impact on detection with fixed routing and OR experts can compare routing strategies on a shared risk map.

**Framework overview.** The benchmarking dataset is organized into **layouts**, each representing a distinct geographic map. Each layout is associated with multiple wildfire **scenarios**, defined as spatiotemporal grids $\mathcal{I} \times \mathcal{T}$ that capture wildfire spread over time along with environmental conditions such as weather and vegetation. Scenarios provide the spatial and temporal context in which detection strategies—covering both device placement and drone routing—are evaluated.

To ensure consistency between simulation dynamics and decision-making, we align each scenario's temporal structure with drones' operational capabilities by distinguishing between two levels of resolution: **data** and **operational**. The *data temporal resolution* defines the duration of each simulation step (e.g., one hour), while the *data spatial resolution* specifies the physical size of each grid point in $\mathcal{I}$. In contrast, the **operational resolution** defines the granularity of decisions. The **operational spatial resolution** corresponds to the distance unit for drone routing, typically a multiple of the drone's coverage radius. The **operational time resolution** indicates how many operational grid cells a drone can traverse per simulation step.

A **strategy** refers to a decision-making policy for wildfire detection, comprising sensor placement and drone routing. The sensor placement component determines where ground sensors and charging stations are deployed, while the drone routing component governs the movement of drones across the grid. Each strategy operates on a **wildfire risk map** where $r_{it} \in [0,1]$ represents the burn probability of grid point $i$ at time $t$. Our library distinguishes between two types of risk maps: *static* (fixed over time) and *dynamic* (continuously updated in response to changing weather or environmental conditions). During runtime, the risk map is continuously adjusted, lowering ignition and burn probabilities in regions currently monitored by drones to model the mitigating impact of real-time aerial surveillance.

Drones operate under physical and operational constraints. These include finite battery life ($D_{\max}$), limited transmission range ($R$), and bounded mobility. The **maximum effective speed** of a drone does not reflect raw flight velocity but instead the rate at which it can effectively scan and update grid points within its coverage area. These constraints, together with the scenario resolution, define the feasible range of strategic decisions.

## 4.1 BENCHMARK DATASETS

**Data collection.** To model wildfire spread, we use the Sim2Real-Fire dataset (22), which contains over one million high-fidelity simulated fire trajectories and 1,000 real-world wildfire cases, each annotated with five aligned modalities: topography, vegetation, fuel, weather, and satellite imagery. While rich in fire dynamics, its ignition points are synthetically and uniformly distributed, limiting realism. To address this, we incorporate historical ignition data from the Fire Program Analysis Fire-Occurrence Database (FPA FOD) (30), which includes 2.17 million geo-referenced ignition records across the U.S. This integration enables more realistic study of ignition and spread behavior while avoiding confounding effects from suppression, using counterfactual simulations from physics-based and empirical models. We constrain matches to ignition points within 150 meters of historical locations and allow any calendar day, further limiting selection to large-scale regions at least 15

| Input | Optimization framework | | | Output |
|---|---|---|---|---|

Figure 1: Architectural diagram showing the benchmark components and how they relate to one another.

kilometers wide for operational relevance. The final curated dataset includes 7746 fire-spread scenarios across 49 rectangular layouts, with final dimensions ranging from 30 km to 60 km.

**Benchmarking dataset categorization.** To enable structured analysis of drone deployment strategies under diverse wildfire conditions, we categorize each fire-spread scenario along five key axes: (i) fire size (small vs. large, with large defined as a final burn radius exceeding 20 km), (ii) spread rate (slow vs. fast, where fast denotes rapid expansion within the first 10 hours), (iii) historical match (alignment in both spatial location and calendar day with recorded wildfire events), (iv) seasonal match (correct region and seasonality), and (v) layout inclusion (all fires occurring within a given deployment layout). This categorization facilitates the evaluation of risk maps and response strategies across a range of fire behaviors and temporal conditions, and is especially important for assessing robustness to both localized and large-scale events (e.g., 20–50 km fires)(36). To avoid temporal bias from fixed ignition times, we introduce a random delay of one to twelve hours in each scenario. This ensures that drone systems begin in a stable operational state and prevents routing algorithms from exploiting prior knowledge of fire onset time.

## 4.2 Evaluation Metrics

To evaluate monitoring strategies, we measure detection time, coverage, and computational cost. The main detection metric is the **mean detection time** or the expected detection delay $\mathbb{E}[\Delta t] = \frac{1}{|\mathcal{W}|} \sum_{(i,t_0) \in \mathcal{W}} (t_d(i) - t_0(i))$, averaged over wildfire scenarios $\mathcal{W}$ per each layout. To focus on early detection, we consider a fire undetected if it is not spotted within its first twelve hours after ignition. We report the percentage of detected fires, the mean fire size at detection, and the proportion of detections attributed to each device type (ground sensors, charging stations, and drones). Spatial efficiency and resource metrics include the mean percentage of the map explored, mean distance traveled, and mean distance to the nearest charging station. To account for algorithmic efficiency, we report the mean drone routing execution time, measuring computational overhead. These metrics provide a comprehensive basis for evaluating trade-offs between responsiveness, resource utilization, and computational cost (see Appendix B for full definitions of evaluation metrics).

## 5 Experiments

We conducted a comprehensive set of experiments to evaluate wildfire detection strategies under realistic, operational constraints. Our aim was to assess the performance of various sensor placement and drone routing methods across diverse terrains and ignition patterns. This section outlines the detection strategies and baselines, wildfire risk map selection, drone specs, and experimental setup, including hardware configurations.

## 5.1 Wildfire Detection Strategies and Baselines

We implement several wildfire detection strategies, consisting of the placement of ground sensors and charging stations, and the routing of drones, and systematically evaluate their performance. To

ensure scalability, charging station locations serve as anchors for a decomposition algorithm (see Appendix C) that partitions the surveillance area into operational zones, each assigned to a subset of drones. An overview of the framework is shown in Figure 1.

**Ground Sensor and Charging Station Placement Baselines**

1. **Gaussian-weighted Max-Coverage** (GaussianCov): We model the placement of ground sensors and drone charging stations as a maximum risk coverage location problem, first introduced by (6). We extend the objective to maximize the total monitored wildfire risk, i.e.,

$$\max_{(\boldsymbol{x^g}, \boldsymbol{x^c}) \in \mathcal{F}_1} \sum_{t \leq H} \sum_{i \in \mathcal{I}} r_{it}^d \dot{\min}(1, K_i(x_i^c), x_i^g)$$

with $\mathcal{F}_1 = \{(\boldsymbol{x^g}, \boldsymbol{x^c}) \in \{0,1\}^{\mathcal{I}} \times \{0,1\}^{\mathcal{I}} \mid x_i^g = 0 \ \forall i \in \mathcal{I} \setminus \mathcal{I}_g \text{ and } x_i^c = 0 \ \forall i \in \mathcal{I} \setminus \mathcal{I}_c\}$ the set of all valid decisions, $r_{it}^d$ the wildfire risk at cell $i$ and time $t$, and $H$ the optimization horizon. Our approach extends traditional max coverage by introducing a kernel function $K_i$ that captures both direct and indirect coverage effects. For ground sensors we consider standard binary coverage of their immediate location. For charging stations, however, we account for the indirect impact on risk coverage through the drones that will utilize them. Beyond covering their immediate cell, charging stations enable drone operations that can monitor surrounding areas. To model this accessibility, we compute $K_i(x_i^c)$ as the probability that a drone starting from charging station locations will visit cell $i$, based on a 2D Brownian motion model. This yields a Gaussian distribution centered at each station, with variance determined by drone battery life. Overlapping coverage is handled by summing individual probabilities, capped at 1. We estimate $K_i$ via iterative $3 \times 3$ convolutions with a uniform kernel to a grid initialized with zeros and ones to simulate diffusion (see Appendix E.1 for details).

2. **Uniform** (Random): We draw uniformly at random the location of each ground sensor and charging station across the grid. This approach ignores wildfire risk, spatial structure, and feasibility constraints. While impractical for real-world use, it provides a useful baseline to assess the benefits of informed, risk-aware placement strategies.

**Drone Routing Baselines**

1. **Max-Coverage** (MaxCov): We formulate drone routing as a rolling-horizon, risk-weighted maximum unique coverage optimization problem. The objective is to maximize cumulative risk observed across all drones and time steps within a planning horizon: $\max_{(\boldsymbol{a},\boldsymbol{b},\boldsymbol{c},\boldsymbol{\theta}) \in \mathcal{F}_2} \sum_{i \in \mathcal{I}_d} \left\{ r_{i1}^d \theta_{i1} + \sum_{t \in \mathcal{H} \setminus \{1\}} r_{it}^d (\theta_{it} - \theta_{i,t-1}) \right\}$, where $\mathcal{F}_2$ represents the set of feasible decisions subject to drone dynamics and operational constraints (formally defined in Appendix E.2). Wildfire risk scores $r_{it}^d$ are updated every $T_{\text{reev}}$ steps based on drone observations and external data sources. This strategy plans over a finite optimization horizon $\mathcal{H}$ to identify routing decisions that maximize first-time surveillance of high-risk areas, while accounting for long-term feasibility, including battery depletion, and station returns.

2. **TOP** (TOP): Our drone routing strategy extends the classical TOP, which itself generalizes TSP. While TSP seeks to visit all locations while minimizing travel cost, TOP allows selecting a subset of locations to visit in order to maximize the sum of profits collected, subject to a travel cost constraint. In our wildfire monitoring context, we treat wildfire risk as the reward, incentivizing drones to visit high-risk areas. Our extension modifies the standard TOP formulation in two key ways: first, rather than requiring drones to start and end at the same depot, our multi-drone system allows each drone to freely choose different charging stations for departure and arrival; second, we employ a rolling optimization approach that optimizes one battery cycle at a time. During each cycle, drones start fully charged from their current charging station locations, jointly optimize their routes to maximize total risk coverage, and must return to a charging station before battery depletion. In subsequent cycles, each drone begins from the charging station where it ended the previous cycle. The total travel distance for each drone is constrained by its battery capacity. We solve this multi-drone joint optimization problem using an adapted particle swarm optimization (PSO) algorithm to improve performance on grid graphs, particularly by adapting the local search algorithm to exploit the grid structure more efficiently (achieving up to 400 times speedup). The detailed problem formulation and solution methodology is provided in Appendix E.3.

3. **Uniform Coverage** (UniCov): As an "uninformed" stochastic baseline, drones are directed to locations sampled uniformly at random within their assigned operational zone. We consider a drone routing strategy for maximum coverage under a uniform risk map, where every grid cell is equally important. This approach, which ignores wildfire risk and prior coverage, serves as a baseline to quantify the added value of risk maps in drone routing.

4. **Brownian** (Brownian): At each step, the next waypoint is sampled with probability proportional to the current risk level. Drones are initially placed uniformly across charging stations and move randomly to neighboring grid points at each time step. All movements are constrained to ensure feasibility with respect to battery levels and charging station accessibility. This strategy ignores wildfire risk and provides a simple, constraint-satisfying baseline.

## 5.2 Wildfire Risk Map Specification

In our experimental setup, we incorporate a real-world, static wildfire Burn Probability (BP) risk map developed by the USDA Forest Service for all U.S. lands (13). BP represents the annual probability of wildfire occurrence at a specific location. The dataset provides high spatial resolution at 30 meters, enabling fine-grained, risk-aware planning for both ground sensor placement and drone routing. It is widely adopted in federal and community-level wildfire risk assessments and is publicly available alongside comprehensive methodological documentation.

Our library also supports dynamic wildfire risk maps that evolve over time. To demonstrate this capability and establish a benchmark for evaluation, we compute a mock "ground-truth" burn map using a frequentist approach. For each layout, we average the simulated fire propagation scenarios over time to estimate the empirical burn probability of each cell at each time step, corresponding to $r_{i,t}^d$ in our framework. This dynamic map enables evaluation of detection strategies under ideal conditions and allows comparison with the static BP map to assess sensitivity to risk map accuracy.

## 5.3 Drone Specification

We model wildfire monitoring using real-world drone specifications to ensure operational feasibility. Our experiments use the Mugin EV350, a long-range, fully electric VTOL drone designed for autonomous environmental monitoring. It offers a 130 km flight range, 120-minute flight time, and a cruising speed of 24 m/s, making it suitable for rapid response over large areas. It supports a 2 kg payload, has a transmission range of up to 80 km, and withstands wind conditions up to Grade 6 (23).

## 5.4 Experimental Setup

To evaluate the effectiveness of our sensor placement and drone routing strategies under realistic wildfire conditions, we design a suite of experiments grounded in operational constraints and historical fire data. Each scenario deploys **eight** ground stations, **two** drones, and **two** charging stations to simulate a realistic and resource-constrained wildfire monitoring operation. To ensure consistency and comparability, all experiments use standardized specifications for the drone platform, charging infrastructure, and ground sensors (see Appendix A for full list of variables and Appendix D for their specific values). For testing, we use a subset of our dataset (introduced in Section 4.1), which has 80% or more historical ignition and simulation (Sim2Real) matches, resulting in 12 layouts with 474 wildfire spread scenarios in total. We adopt a high match threshold of 80% to improve the alignment with real-world risk patterns and to maintain statistical consistency across scenarios.

Our strategies (introduced in Section 5.1) comply with operational constraints of planning drone routing in one-hour increments. The runtime remains efficient across methods. For the Max-Coverage, Uniform, and Brownian strategies, each hour of operation requires less than 30 seconds of computation. TOP is designed to run in real time, taking approximately one hour of computation for each hour of operation. Ground station placement strategies require less than two seconds.

**Hardware specification.** All our experiments are conducted in parallel on a cluster of Intel Xeon Platinum 8260. Each burn map and sensor placement strategy combination is run as a single job on 32 CPUs with 125 GB of RAM. Within each job, the drone strategies are run sequentially. For each drone strategy, every layout is run in parallel on a separate thread. The routing and placement

strategies are implemented in Julia version 1.11.3 using the `JuMP.jl` version 1.26.0 optimization modeling package and solved with Gurobi version 12.0.3 (17).

## 6 RESULTS AND DISCUSSION

Our experimental results using a real-world wildfire BP map are shown in Table 2. To illustrate the outcomes under an ideal risk map, assess sensitivity to risk map quality, and demonstrate the use of the dynamic risk map available in our library, we present results using our "ground truth" risk map (described in Section 5.2) in Table 3. This map serves as a high-fidelity benchmark with hourly resolution, enabling us to evaluate the upper bound of performance for risk-informed strategies. These results reveal several key insights.

Most notably, the best strategy depends on risk-map fidelity – risk-aware approaches excel when risk maps are accurate, whereas spread-out coverage performs better under a less informative risk map. From Table 2, we observe that Uniform Coverage frequently yields the highest detection rates – and even achieves perfect detection (100%) in all fast-fire scenarios – while maintaining reasonable detection times. By assuming all areas have the same risk, Uniform Coverage avoids the risk of missing fires due to inaccurate risk assessments. In contrast, risk-based routing strategies such as TOP and Max-Coverage can detect a subset of fires very quickly (e.g., 2.00–3.40 hours detection in Fast Big) when the risk map is accurate, but they miss many fires when the BP risk prediction provides unreliable guidance (e.g., 45% and 73% versus 100% detection in Fast Big). Importantly, Table 3 shows the opposite behavior under the ground-truth map: when risk information is perfect, TOP and Max-Coverage outperform Uniform Coverage on detection speed while performing similarly on overall detection rate. This demonstrates that risk-aware routing can be substantially more efficient when risk maps are sufficiently accurate, as Uniform coverage cannot exploit the efficiency gains that come from focusing on high-risk regions. The dependence on risk-map fidelity highlights a central challenge in wildfire monitoring: optimizing drone routing requires matching the routing strategy to the quality of risk information available.

Second, the comparison between Max-Coverage and Uniform Coverage further highlights the importance of the risk map. Since Uniform Coverage is an ablation of Max-Coverage that follows the same routing logic but assumes a uniform risk distribution, differences in their performance directly reflect how effectively Max-Coverage leverages the input risk information. When using the dynamic ground-truth map, Max-Coverage consistently outperforms Uniform Coverage in terms of detection time. However, this advantage largely disappears when the static BP risk map is used. This trend demonstrates the critical importance of accurate, temporally dynamic risk maps in supporting the effective operation of risk-informed strategies.

Third, risk-aware static infrastructure helps even under an imperfect risk map, and becomes even more beneficial when the map is accurate. This can be observed from the consistent improvement of GaussianCov over Random sensor placement in both Table 2 and 3 as in nearly all fire types, GaussianCov yields higher detection rates and often faster detection times because the ground sensors and charging stations are concentrated around regions identified as high risk.

Fourth, drone-based detection consistently outperforms ground sensors, with drones identifying most fires despite only two drones versus eight sensors and two charging stations (Appendix F). This suggests drones may offer better returns on investment due to their greater coverage.

Finally, unsurprisingly, the Brownian strategy is the worst-performing baseline.

**Limitations.** Despite its strengths, `WFDroneBench` has several limitations. The simulated environment abstracts away operational complexities such as communication failures, terrain constraints, and execution uncertainties. Additionally, the current implementation assumes full observability and reliable risk updates, which may not hold in real deployments. Finally, environmental factors (e.g., wind, temperature) that may directly impact drone performance are not considered. Since such factors typically vary little within an hour, they can be incorporated into the rolling-horizon framework without jeopardizing the drones' ability to return to a charging station, as long as the optimization horizon is small enough.

**Future research.** While `WFDroneBench` currently emphasizes optimization-based routing, reinforcement learning (RL) (11) presents a promising avenue for adaptive decision-making in complex,

| | | Overall | Slow Small | Slow Big | Fast Small | Fast Big |
|---|---|---|---|---|---|---|
| **Sensor Strategy** | **Drone Strategy** | | | | | |
| Random | TOP | 80%; **2.96** (5.29) | 63%; **2.50** (4.37) | 81%; 4.08 (5.53) | 86%; **1.67** (3.61) | 82%; **7.45** (7.46) |
| | MaxCov | 91%; 3.19 (4.42) | 78%; 3.75 (3.40) | 93%; 3.54 (4.40) | 100%; 2.29 (3.40) | 55%; 9.67 (9.63) |
| | UniCov | **93%**; 3.08 (4.23) | **88%**; 3.97 (4.84) | **94%**; **3.21** (4.12) | **100%**; 3.00 (2.08) | **100%**; 7.45 (5.35) |
| | Brownian | 74%; 5.22 (6.71) | 46%; 5.37 (6.99) | 77%; 7.66 (6.52) | 57%; 5.25 (4.72) | 82%; 8.78 (6.28) |
| GaussianCov | TOP | 84%; **3.16** (4.76) | 76%; **3.71** (4.42) | 85%; 4.06 (4.92) | 86%; 4.67 (3.72) | 45%; 3.40 (2.19) |
| | MaxCov | 96%; **3.16** (4.24) | 90%; 5.11 (5.65) | **97%**; 3.23 (4.08) | **100%**; 3.71 (6.90) | 73%; **2.00** (2.14) |
| | UniCov | **97%**; 3.17 (4.17) | **93%**; 5.05 (5.45) | **97%**; **3.18** (3.97) | **100%**; **1.00** (1.00) | **100%**; 5.73 (6.17) |
| | Brownian | 81%; 4.99 (6.33) | 59%; 6.96 (7.42) | 83%; 6.34 (6.31) | 86%; 8.00 (4.69) | 91%; 6.10 (3.51) |

Table 2: Percentage of fires detected in under 12 hours and mean detection time (±SD in hours) using the BP risk map, across sensor/drone strategies and fire scenarios (by size and speed).

| | | Overall | Slow Small | Slow Big | Fast Small | Fast Big |
|---|---|---|---|---|---|---|
| **Sensor Strategy** | **Drone Strategy** | | | | | |
| Random | TOP | **92%**; 2.69 (4.03) | 76%; **2.68** (3.81) | **94%**; 3.04 (4.08) | 86%; 2.50 (4.32) | **100%**; **4.18** (4.38) |
| | MaxCov | 89%; **2.49** (3.74) | 78%; 3.72 (4.39) | 89%; **2.88** (3.66) | **100%**; 2.14 (3.67) | **100%**; 4.55 (5.24) |
| | UniCov | **92%**; 3.74 (5.11) | **93%**; 4.71 (5.04) | 92%; 4.19 (5.22) | 91%; **1.86** (4.06) | 91%; 4.20 (4.05) |
| | Brownian | 66%; 3.77 (5.55) | 29%; 7.83 (4.73) | 68%; 6.16 (5.38) | 43%; 7.67 (1.15) | 91%; 5.40 (5.93) |
| GaussianCov | TOP | 91%; **3.04** (4.44) | 73%; **2.80** (4.01) | 92%; 3.49 (4.49) | **100%**; **0.86** (1.21) | **100%**; 6.45 (5.52) |
| | MaxCov | 95%; 3.20 (4.21) | 78%; 4.44 (6.16) | 96%; **3.32** (4.01) | **100%**; 3.57 (3.21) | **100%**; **5.73** (5.50) |
| | UniCov | **96%**; 3.41 (4.39) | **83%**; 3.35 (3.60) | **97%**; 3.58 (4.40) | **100%**; 2.14 (2.48) | 82%; 8.22 (5.93) |
| | Brownian | 73%; 5.11 (6.53) | 63%; 7.92 (5.93) | 75%; 7.23 (6.34) | 86%; 9.00 (7.13) | 36%; 8.75 (5.32) |

Table 3: Percentage of fires detected in under 12 hours and mean detection time (±SD in hours) using the "ground truth" dynamic map, across sensor/drone strategies and fire scenarios (by size and speed).

uncertain environments. RL methods could enable end-to-end policies that respond directly to environmental feedback and stochastic fire behavior, potentially reducing dependence on precise real-time risk inputs. Other directions for future exploration include incorporating high-resolution, weather-conditioned dynamic risk maps, designing routing strategies that minimize expected detection time, and investigating heterogeneous fleet deployments. Additional work may also involve integrating dynamic predictions with real-time decision-making and expanding the benchmark suite with datasets derived from historical wildfire propagation events. Additionally, future work could extend the framework's relevance to policy makers and mechanical engineers by enabling deeper analyses of resource-allocation decisions and by quantifying how drone hardware characteristics – such as battery capacity and communication range – shape achievable detection performance, as illustrated in the broader stakeholder landscape of Figure 6 in Appendix J.

Our findings highlight two important open challenges: (i) achieving rapid, reliable detection of small fires, and (ii) improving wildfire risk-map prediction, where the gap between ground-truth ignition patterns and available risk maps underscores a substantial opportunity for machine learning innovation.

**Reproducibility statement.** We are committed to open and reproducible research. All code, data, and experimental configurations used in this study are included as supplementary material and are publicly available and free to access. The README file in the supplementary material provides detailed instructions for reproducing our results, including environment setup and experiment scripts. This enables all experiments to be reliably replicated and extended by the research community.

**Ethics and LLM statement.** An LLM was used to shorten and improve the readability of the text.

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

## A    Summary of sets, parameters, and decision variables

To ensure clarity in the mathematical formulations presented in the appendices, Table 4 summarizes all sets, parameters, and decision variables referenced in the appendices.

Table 4: Overview of sets, parameters, and decision variables used in the supplementary materials.

| Sets | Description | Definition |
|---|---|---|
| $\mathcal{I}$ | Set of all possible grid points | |
| $\mathcal{I}_g$ | Feasible grid points for ground stations | $\subseteq \mathcal{I}$ |
| $\mathcal{I}_c$ | Feasible grid points for charging stations | $\subseteq \mathcal{I}$ |
| $\mathcal{I}_d$ | Feasible grid points for drones | $\subseteq \mathcal{I}$ |
| $\mathcal{A}_d(i)$ | Neighboring grid points of $i$ within distance $d$ (including $i$) | |
| $\mathcal{T}$ | Set of all time periods | $\{1, \ldots, T\}$ |
| $\mathcal{H}$ | Time periods in optimization time horizon | $\{1, \ldots, H\}$ |
| $\mathcal{S}$ | Set of drones | $\{1, \ldots, S\}$ |
| $\mathcal{C}$ | Set of charging stations | Cardinality $C$ |
| $\mathcal{G}$ | Set of ground sensors | Cardinality $G$ |
| $\mathcal{N}_c$ | Nearby charging station pairs | $\{(i,j) \in \mathcal{I}_c^2 \mid i \neq j, \left\|i - j\right\| \leq \rho_c\}$ |
| $\mathcal{N}_g$ | Nearby ground sensor pairs | $\{(i,j) \in \mathcal{I}_g^2 \mid i \neq j, \left\|i - j\right\| \leq \rho_g\}$ |
| $\mathcal{N}_{cg}$ | Charging–ground sensor pairs within exclusion radius | $\{(i,j) \in \mathcal{I}_c \times I_g \mid i \neq j, \left\|i - j\right\| \leq \rho_{cg}\}$ |
| $\mathcal{W}$ | Set of historical wildfires | |

| Parameters | Description |
|---|---|
| $\Delta_i$ | Minimum distance from grid point $i$ to nearest charging station |
| $B_{\max}$ | Maximum number of time steps a drone can fly without recharging |
| $r_i^s$ | Static wildfire risk at grid point $i$ |
| $r_{it}^d$ | Dynamic wildfire risk at grid point $i$ at time $t$ |
| $f_{it}^w$ | Indicator of if grid point $i$ is burnt at time $t$ in wildfire scenario $w$. |
| $n_g$ | Maximum number of ground sensors that can be placed |
| $n_c$ | Maximum number of charging stations that can be placed |
| $n_d$ | Maximum number of drones assignable to each charging station |
| $\tau$ | Coverage memory window: revisits provide no reward during this time |
| $\rho_c$ | Exclusion radius: minimum spacing between charging stations |
| $\rho_g$ | Exclusion radius: minimum spacing between ground sensors |
| $\rho_{cg}$ | Exclusion radius: minimum spacing between charging stations and ground sensors |
| $\lambda_{\text{disp}}$ | Weight of drone dispersion in the objective function |
| $t_{\text{det}}^w$ | Time of detection of wildfire $w \in \mathcal{W}$ |
| $t_{\text{ign}}^w$ | Time of ignition of wildfire $w \in \mathcal{W}$ |
| $t_{\text{det}}^s$ | Time at which drone $s$ detects a wildfire; equals $T$ if no fire is detected by drone $s$ |
| $t_{\text{exe}}$ | Execution time of the drone routing algorithm |
| $\boldsymbol{p}^c$ | Grid coordinates in $\mathbb{Z}^2$ of charging station $c$ |
| $T_{\text{reev}}$ | Reevaluation step for rolling optimization |

| Variables | Description |
|---|---|
| $x_i^g$ | Binary variable denoting if a ground sensor is placed at grid point i |
| $x_i^c$ | Binary variable denoting if a charging station is placed at grid point i |
| $a_{its}$ | Binary variable denoting if drone $s$ flies at grid point $i$ at time $t$ |
| $c_{its}$ | Binary variable denoting if drone $s$ charges at grid point $i$ at time $t$ |
| $\theta_{it}$ | Binary variable denoting if grid point $i$ is covered by a drone at time $t$ |
| $b_{ts}$ | Battery of drone $s$ at time $t$ defined as the number of time steps drone $s$ can operate without recharging |
| $\boldsymbol{p}^{s,t}$ | Grid coordinates in $\mathbb{Z}^2$ of drone $s$ at time $t$ |

## B    Definitions of evaluation metrics

In Table 5 we give the mathematical formulations of the evaluation metrics as described in Section 4.2 of the main paper.

| Evaluation metric | Formula |
|---|---|
| Mean detection time (MDT) | $\frac{1}{\sum_{w\in\mathcal{W}}\mathbb{1}\{t_{\mathrm{det}}^w \leq T\}}\sum_{w\in\mathcal{W}}\mathbb{1}\{t_{\mathrm{det}}^w \leq T\}(t_{\mathrm{det}}^w - t_{\mathrm{ign}}^w)$ |
| Mean fire size at detection (MSD) | $\frac{1}{|\mathcal{I}|}\sum_{i\in\mathcal{I}} f_{it}^w$, with $t = t_{\mathrm{exe}}^w$ |
| % fires detected (FDR) | $\frac{1}{|\mathcal{W}|}\sum_{w\in\mathcal{W}}\mathbb{1}\{t_{\mathrm{det}}^w \leq T\} \cdot 100\%$ |
| Proportion of detection by device D (DS-D) | $\frac{\sum_{w\in\mathcal{W}}\mathbb{1}\{t_{\mathrm{det}}^w \leq T\}\mathbb{1}\{\text{fire detected by device D}\}}{\sum_{w\in\mathcal{W}}\mathbb{1}\{t_{\mathrm{det}}^w \leq T\}}$ |
| Mean percentage of map explored (MCE) | $\frac{1}{|\mathcal{I}|}\sum_{i\in\mathcal{I}}\mathbb{1}\left(\sum_{t\in\mathcal{T}}\theta_{it} \geq 1 \vee x_i^g = 1 \vee x_i^c = 1\right)$ |
| Mean distance traveled by drones (MDist) | $\frac{1}{S}\sum_{s\in\mathcal{S}}\sum_{t=1}^{t_{\mathrm{det}}^s} d\left(\boldsymbol{p^{s,t}}, \boldsymbol{p^{s,t+1}}\right)$ |
| Mean distance to nearest charging station (DCSD) | $\frac{1}{ST}\sum_{s\in\mathcal{S}}\sum_{t\in\mathcal{T}}\min_{c\in\mathcal{C}} d\left(\boldsymbol{p^{s,t}}, \boldsymbol{p^c}\right)$ |
| Mean drone routing execution time (RTE) | $\frac{1}{\#\mathrm{scenarios}}\sum_{i=1}^{\#\mathrm{scenarios}} t_{\mathrm{exe}}$ |

Table 5: Mathematical formulations of the evaluation metrics.

## C  DECOMPOSITION ALGORITHM

To enhance scalability and reduce computational complexity, we partition the full set of charging stations into spatial operational zones, where each operational zone includes stations that are mutually reachable within a drone's battery range. Specifically, two stations are placed in the same operational zone if the Euclidean distance between them is less than or equal to the drone's battery range, ensuring that any drone can travel between stations in the operational zone. The decomposition algorithm is given in Algorithm 1.

---

**Algorithm 1** Decomposition model

---

**Input**: set of charging stations $\mathcal{C}$.
**Output**: operational zones $\mathcal{K}_1, \ldots, \mathcal{K}_m$, where $\mathcal{K}_k \subseteq \mathcal{C}$ for all $k \in [m]$, $\cap_{k\in[m]}\mathcal{K}_k = \emptyset$.

1: Set $k = 1$
2: **while** $\mathcal{C} \neq \emptyset$ **do**
3:     Select a charging station $i \in \mathcal{C}$
4:     Remove $i$ from $\mathcal{C}$
5:     Set $\mathcal{K}_k = \{i\}$
6:     **for** $j \in \mathcal{C} \setminus \{i\}$ **do**
7:         **if** the Euclidean distance between $i$ and $j$ is less or equal than $\lambda$ **then**
8:             add $j$ to operational zone $\mathcal{K}_k$
9:             remove $j$ from $\mathcal{C}$
10:       **end if**
11:     **end for**
12: **end while**
13: **return** $\mathcal{K}_1, \ldots, \mathcal{K}_m$.

---

## D  PARAMETER VALUES USED IN THE EXPERIMENTS

In Table 6 we give an overview of the parameter values used in the experiments when using the max-coverage ground sensor placement baseline in combination with a drone routing strategy.

| Parameter | Value |
|---|---|
| $n_g$ | 8 |
| $n_c$ | 2 |
| $n_d$ | 2 |
| $\rho_c$ | 10 |
| $\rho_{cg}$ | 10 |
| $\rho_g$ | 1 |
| Drone speed (m/min) | 600 |
| Coverage radius (m) | 300 |
| Grid point size (m) | 30 |
| Transmission range (m) | 50000 |
| $T_{\text{reev}}$ | 5 |
| $H$ | 10 |
| $B_{\text{max}}$ (h) | 1 |

Table 6: Parameter values used in the experiments.

## E    MATHEMATICAL MODEL FORMULATIONS

In this section the mathematical formulations are given for all models. All distances are measured in the $L^\infty$ norm, unless specified otherwise.

### E.1    GROUND SENSORS AND CHARGING STATIONS PLACEMENT MODEL

The simplest version of the ground sensors and charging stations max-coverage model only considers direct coverage of a device at its immediate location. We first present this model before extending it to the indirect and partial coverage model we used. The simple model is given by:

$$\max_{\boldsymbol{x}^g, \boldsymbol{x}^c} \quad \sum_{i \in \mathcal{I}_g} r_i^s x_i^g + \sum_{i \in \mathcal{I}_c} r_i^s x_i^c \tag{1a}$$

$$\text{s.t.} \quad x_i^g + x_i^c \leq 1, \qquad\qquad \forall i \in \mathcal{I}_g \cap \mathcal{I}_c, \tag{1b}$$

$$\sum_{i \in \mathcal{I}_g} x_i^g \leq n_g, \tag{1c}$$

$$\sum_{i \in \mathcal{I}_c} x_i^c \leq n_c, \tag{1d}$$

$$x_i^g + x_i^c \leq 1, \qquad\qquad (i,j) \in \mathcal{N}_g, \tag{1e}$$

$$x_i^g + x_j^c \leq 1, \qquad\qquad (i,j) \in \mathcal{N}_c, \tag{1f}$$

$$x_j^g + x_i^c \leq 1, \qquad\qquad (i,j) \in \mathcal{N}_{cg}. \tag{1g}$$

Here, constraints (1b) ensure that there can either be a charging station or a ground sensor at grid point $i$, but not both. Constraints (1c) and (1d) are capacity constraints on the ground sensors and charging stations respectively. Finally, (1e)-(1g) are spatial exclusion constraints between two ground sensors, two charging stations, and a ground sensor and a charging station, respectively.

**Extensions.** Observe that we have assumed that a ground sensor as well as a charging station only detects a wildfire at the grid point it occupies. This can be naturally extended to allow partial detection of wildfires in neighboring grid points by replacing (1) by

$$\max_{\boldsymbol{\phi}, \boldsymbol{x}, \boldsymbol{y}} \quad \sum_{i \in \mathcal{I}_g \cup \mathcal{I}_c} \phi_i$$

$$\text{s.t.} \quad \phi_i \leq \sum_{k \in \mathcal{I}_g \cap \mathcal{A}_d(k)} \gamma_{ik}^g x_k^g + \sum_{k \in \mathcal{I}_c \cap \mathcal{A}_{d'}(k)} \gamma_{ik}^c x_k^c, \qquad i \in \mathcal{I}_g \cup \mathcal{I}_c,$$

$$0 \leq \phi_i \leq 1, \qquad\qquad i \in \mathcal{I}_g \cup \mathcal{I}_c,$$

where $\gamma_{ik}^G$ and $\gamma_{ik}^C$ denote the partial contribution from a ground sensor and charging station at grid point $k$ to grid point $i$ respectively, and $\phi_i^g$ denotes the total fractional coverage in grid point $i$ due to ground sensors and charging stations nearby.

Observe that our model readily allows for the integration of existing ground sensors alongside newly optimized placements by explicitly including them in the formulation with fixed variable values (e.g., setting $x_i^g = 1$ for grid points with pre-existing sensors) or alternatively excluding them from the decision variable domains.

**Gaussian Coverage** Notice that the Gaussian coverage model we use and presented in section 5.1 is a special case of the extension presented above with the following parameters:

- $\gamma_{ik}^g$ is 1 for $i = k$ and 0 otherwise, as ground sensors only cover their immediate location.
- All $\gamma_{ik}^c$ are precomputed at the same time for a given $i$. We start with a grid of zeros with a unique 1 at cell $i$ and iteratively take convolutions with a $3 \times 3$ kernel of ones, simulating the diffusion process of a drone over neighboring cells. After dividing the resulting grid by the number of convolutions applied, the value of cell $k$ gives the expected number of visits of cell $k$ by a drone following a Brownian motion starting at $i$. Capping this value by 1 gives $\gamma_{ik}^c$.

Expressing the Gaussian coverage as a special case of the extended max coverage model allows us to solve it efficiently using linear programming.

### E.2    MAX COVERAGE DRONE ROUTING MODEL

The drone routing optimization model with the objective to maximize coverage is given by:

$$\max_{\boldsymbol{a},\boldsymbol{b},\boldsymbol{c},\boldsymbol{\theta}} \quad \sum_{i \in \mathcal{I}_d} \left\{ r_{i1}^d \theta_{i1} + \sum_{t \in \mathcal{H}\setminus\{1\}} r_{it}^d (\theta_{it} - \theta_{i,t-1}) \right\} \tag{2a}$$

$$\text{s.t.} \quad \sum_{i \in \mathcal{I}_d} a_{its} + \sum_{i \in \mathcal{C}} c_{its} = 1, \qquad\qquad t \in \mathcal{H} \setminus \{1\}, s \in \mathcal{S}, \tag{2b}$$

$$\sum_{s \in \mathcal{S}} c_{its} \leq n_d, \qquad\qquad i \in \mathcal{C}, t \in \mathcal{H} \setminus \{1\}, \tag{2c}$$

$$c_{j,t+1,s} + a_{j,t+1,s} \leq \sum_{i \in \mathcal{I}_d \cap \mathcal{A}_1(j)} a_{its} + c_{jts}, \qquad j \in \mathcal{C}, t \in \mathcal{H} \setminus \{H\}, s \in \mathcal{S}, \tag{2d}$$

$$a_{j,t+1,s} \leq \sum_{i \in \mathcal{I}_d \cap \mathcal{A}_1(j)} a_{its}, \qquad\qquad j \in \mathcal{I}_d \setminus \mathcal{C}, t \in \mathcal{H} \setminus \{H\}, s \in \mathcal{S}, \tag{2e}$$

$$0 \leq b_{ts} \leq B_{\max}, \qquad\qquad t \in \mathcal{H}, s \in \mathcal{S}, \tag{2f}$$

$$b_{ts} \geq B_{\max} \sum_{i \in \mathcal{C}} c_{its}, \qquad\qquad t \in \mathcal{H}, s \in \mathcal{S}, \tag{2g}$$

$$b_{t+1,s} \leq b_{ts} - 1 + (B_{\max} + 1) \sum_{j \in \mathcal{C}} c_{j,t+1,s} \qquad t \in \mathcal{H} \setminus \{H\}, s \in \mathcal{S}, \tag{2h}$$

$$b_{Ts} \geq D_i \cdot a_{iTs}, \qquad\qquad i \in \mathcal{I}_d, s \in \mathcal{S}, \tag{2i}$$

$$\theta_{it} = 0, \qquad\qquad i \in \mathcal{I}_d \cap (\mathcal{G} \cup \mathcal{C}), t \in \mathcal{H}, \tag{2j}$$

$$\theta_{it} \geq a_{its}, \qquad\qquad i \in \mathcal{I}_d, t \in \mathcal{T}, s \in \mathcal{S}, \tag{2k}$$

$$\theta_{1t} \leq \sum_{s \in \mathcal{S}} a_{1ts}, \qquad\qquad t \in \mathcal{H}, \tag{2l}$$

$$\theta_{it} \leq \sum_{s \in \mathcal{S}} a_{its} + \theta_{i,t-1}, \qquad\qquad i \in \mathcal{I}_d, t \in \mathcal{H} \setminus \{1\}, \tag{2m}$$

$$\theta_{it} \geq \theta_{i,t-1}, \qquad\qquad i \in \mathcal{I}_d, t \in \mathcal{H} \setminus \{1\}. \tag{2n}$$

Constraints (2b) ensure that each drone at time $t$ is either charging or flying, but not both. Constraints (2c) ensure that there can be at most $n_d$ drones at a time at a charging station. Constraints (2d) - (2e) ensure that a drone can only fly or charge at location $j$ at time $t + 1$ if it was charging already in the same location or the drone was in a neighboring location at time $t$. (2f) - (2h) are battery constraints that ensure that when a drone is charging, the battery updates to $B_{\max}$, when a drone is flying, the

battery decreases with 1 every time step, and the battery does not drop below 0. Constraints (2i) enforce a "no-suicide" condition for drones at the end of the planning horizon, that is, a drone must have enough battery remaining at the final time step to reach the nearest charging station from its final position. Constraints (2j)-(2n) are coverage constraints. Constraints (2j) make sure that there is no need to consider drone-based coverage at points where ground sensors or charging stations are already installed. Constraints (2k) ensures that grid point $i$ must be covered at time $t$ when there is a drone present at grid point $i$ at time $t$. Constraints (2l) and (2m) ensure that a grid point is only covered at time $t$ if at least one drone visits it at time $t$ or it was already covered at time $t-1$. Constraints (2n) ensure that once a grid point is detected, it remains detected in all future time steps.

The drone routing is performed using a rolling-horizon optimization framework (see Figure 2). At each decision step, we solve the above MILP over a limited optimization horizon to account for the future implications of routing decisions while maintaining tractability. Only the immediate $T_{\text{reev}}$ steps (depicted in purple in Figure 2) are executed before risk data is updated and the optimization is re-solved. This enables the system to adapt dynamically to evolving wildfire conditions based on drone observations (e.g., risk decay following surveillance) or external updates such as weather changes.

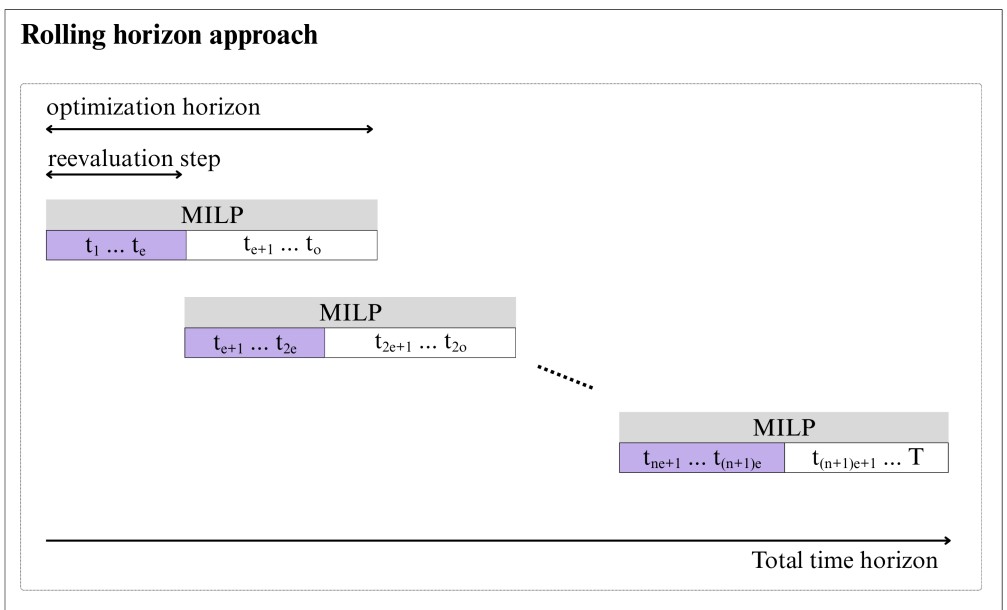

Figure 2: The rolling horizon optimization framework. The problem is solved over an optimization horizon of $o$ time steps using Mixed Integer Linear Programming (MILP). Even though solutions are generated for $o$ time steps, we only use the $e$ first ones, with $e$ the reevalution step parameter. After each reevaluation step of $e$ time steps, the horizon is shifted forward, and the problem is solved again. This process continues until the end of the total time horizon is reached, ensuring adaptability to new information and improved solution quality over time.

In addition, we initialize the drone routing model by adding the following constraints to the optimization problem formulation

$$\sum_{i \in \mathcal{C}} a_{i1s} + \sum_{i \in \mathcal{C}} c_{i1s} = 1, \qquad\qquad s \in \mathcal{S}, \qquad\qquad (3)$$

$$\sum_{s \in \mathcal{S}} (a_{i1s} + c_{i1s}) \leq n_d, \qquad\qquad i \in \mathcal{C}, \qquad\qquad (4)$$

$$b_{1s} = B_{\max}, \qquad\qquad s \in S, \qquad\qquad (5)$$

ensuring that all drones start flying from a charging station (constraints (3)), there are at most $n_d$ drones either flying or charging at a charging station at $t = 1$ (constraints (4)), and each drone starts with full capacity (constraints (5)).

**Extensions.** Observe that we have assumed that a drone only detects a wildfire at the grid point it occupies. This can be naturally extended to allow partial detection of wildfires in neighboring grid points by replacing constraint (2k) by

$$\theta_{it} \leq \sum_{s \in S} \sum_{k \in \mathcal{I}_d \cap \mathcal{A}_d(i)} \gamma_{ik}^d a_{kts}, \qquad\qquad i \in \mathcal{I}_d, t \in \mathcal{T},$$

where $\gamma_{ik}^d$ denotes the partial contribution from a drone at grid point $k$ to grid point $i$. In this case, the variable $\theta_{it}$ may take fractional values and is interpreted as the degree of coverage at location $i$ and time $t$.

To encourage drones to spatially disperse we can include a separation reward term in the objective, defined as the sum of pairwise $L_\infty$ distances between drones over time, i.e., we consider (2) in which we replace the objective (2a) by

$$\max \left\{ \sum_{i \in \mathcal{I}_d} \sum_{t \in \mathcal{T}} r_{it}\phi_{it} + \lambda_{\text{disp}} \sum_{t \in \mathcal{T}} \sum_{s_1 \in \mathcal{S}} \sum_{s_2 < s_1, s_2 \in \mathcal{S}} \max \left\{ \left| p_1^{s_1,t} - p_1^{s_2,t} \right|, \left| p_2^{s_1,t} - p_2^{s_2,t} \right| \right\} \right\}. \quad (6)$$

Note that we can easily linearize the objective (6) using auxiliary variables and big-M constraints.

### E.3 TOP DRONE ROUTING MODEL

Our drone routing strategy extends the classical TOP to accommodate the specific requirements of wildfire monitoring with multiple drones and flexible charging station usage. This section provides the detailed mathematical formulation and solution methodology.

#### E.3.1 GRAPH-BASED TOP FORMULATION

Following the formulation in (8), we model TOP with a graph $G = (V \cup \{d\} \cup \{a\}, E)$, where $V = \{1, 2, \ldots, n\}$ is the set of vertices representing grid cells (customers in the original TOP terminology), $E = \{(i,j) | i, j \in V, i \text{ and } j \text{ neighbors}\}$ is the edge set connecting adjacent grid cells, and $d$ and $a$ are respectively departure and arrival vertices for drones. Each vertex $i \in V$ is associated with a profit $P_i$ (corresponding to wildfire risk $r_i$ in our context), and each edge $(i,j) \in E$ is associated with a constant travel cost.

A tour $R$ is represented as an ordered list of $q$ customers from $V$, so $R = (R[1], \ldots, R[q])$. Each tour begins at the departure vertex and ends at the arrival vertex. The total profit collected from a tour $R$ is $P(R) = \sum_{i=1}^{q} P_{R[i]}$, and the total travel cost/time is $C(R) = C_{d,R[1]} + \sum_{i=1}^{q-1} C_{R[i],R[i+1]} + C_{R[q],a}$. A tour $R$ is feasible if $C(R) \leq L$ with $L$ being the predefined travel cost/time limit (battery capacity in our case).

The fleet is composed of $m$ identical vehicles (drones). A solution $S$ is consequently a set of $m$ (or fewer) feasible tours in which each customer is visited at most once. The goal is to find a solution $S$ such that $\sum_{R \in S} P(R)$ is maximized. In our case, we use wildfire risk as the profit.

#### E.3.2 MODIFICATIONS FOR MULTI-DRONE FLEXIBLE ROUTING

Our approach differs from standard TOP in two key aspects. First, drones can start and end at different charging stations, unlike traditional TOP where vehicles return to their origin depot. Second, in subsequent battery cycles, each drone must start from the charging station where it ended the previous cycle.

To handle these modifications, we introduce a dummy depot node connected to dummy charging station nodes. The number of dummy nodes for each charging station equals the maximum number of drones allowed to start or end at that station. By using the same number of nodes as available drones, we can ensure that drones start at specific charging stations in subsequent cycles. The dummy depot has zero risk/profit, consistent with standard TOP depot formulations and is the starting and ending point of each drone.

### E.3.3 PARTICLE SWARM OPTIMIZATION SOLUTION

We solve this multi-drone TOP using an adapted Particle Swarm Optimization (PSO) algorithm based on (8). Our implementation includes several modifications to improve performance on grid graphs, particularly by adapting the local search algorithm to exploit the grid structure more efficiently.

**Warm-start heuristic:** We initialize particles with high-quality solutions using a greedy heuristic applied to as many particles as we have drones. The heuristic operates as follows:

1. The drone selects the highest-risk cell it can reach given its remaining battery, considering the return trip to any charging station.

2. To reach the selected cell, the drone follows the highest-risk path among all shortest paths to that destination, computed using dynamic programming.

3. Steps 1-2 repeat until the drone's remaining battery equals the distance to the nearest charging station, at which point it returns.

**Optimization process:** The PSO algorithm iteratively refines solutions through particle interactions and local search. We impose a computational time limit of 1 hour for planning each hour of drone operations to balance solution quality with operational constraints.

**Solution extension:** If the PSO converges to a solution shorter than the maximum battery life, we extend it using the same greedy heuristic from the drone's current ending position. This ensures full utilization of available battery capacity when the optimization algorithm terminates early.

## F RESULTS OF REMAINING EVALUATION METRICS

The results across strategies and layouts for each secondary metric are displayed in Table 7 and Table 8.

| sensor strategy | drone strategy | execution time | fire size cells | fire percentage | map explored | total distance | drone detection pct |
|---|---|---|---|---|---|---|---|
| Random | MaxCov | 20.315 (9.243) | 317.865 (500.740) | 0.115 (0.162) | 0.089 (0.058) | 10811.368 (8586.738) | 99.7% (1.0%) |
| | TOP | 2555.125 (1044.639) | 334.081 (393.133) | 0.128 (0.149) | 0.100 (0.052) | 14058.159 (12273.762) | 99.2% (2.6%) |
| | UniCov | 13.777 (1.405) | 394.954 (659.147) | 0.137 (0.220) | 0.133 (0.051) | 12539.380 (8558.823) | 100.0% (0.0%) |
| | Brownian | 0.668 (0.150) | 1891.451 (2025.248) | 0.818 (0.699) | 0.346 (0.224) | 40051.673 (19738.856) | 80.6% (37.9%) |
| GaussianCov | MaxCov | 19.748 (9.022) | 198.689 (239.398) | 0.076 (0.076) | 0.106 (0.058) | 11170.125 (6990.986) | 94.0% (14.1%) |
| | TOP | 2818.260 (531.631) | 380.725 (481.171) | 0.150 (0.187) | 0.117 (0.049) | 13585.247 (12393.535) | 94.2% (13.3%) |
| | UniCov | 12.263 (1.566) | 295.289 (387.528) | 0.107 (0.127) | 0.120 (0.063) | 14744.375 (13057.346) | 95.2% (10.4%) |
| | Brownian | 0.643 (0.133) | 2011.373 (2886.966) | 0.835 (1.081) | 0.346 (0.213) | 39164.918 (20949.228) | 71.5% (40.2%) |

Table 7: Results with baseline sensor strategies

| sensor strategy | drone strategy | execution time | fire size cells | fire percentage | map explored | total distance | drone detection pct |
|---|---|---|---|---|---|---|---|
| Random | MaxCov | 0.000 (0.000) | 641.130 (1152.963) | 0.225 (0.396) | 0.113 (0.060) | 14930.118 (12428.318) | 99.4% (1.1%) |
| | TOP | 2317.133 (1110.519) | 655.801 (1206.986) | 0.216 (0.377) | 0.096 (0.033) | 14005.731 (14018.440) | 99.9% (0.4%) |
| | UniCov | 0.000 (0.000) | 577.594 (1180.993) | 0.204 (0.407) | 0.132 (0.057) | 14472.535 (11741.451) | 99.5% (1.0%) |
| | Brownian | 0.618 (0.120) | 1094.617 (1523.182) | 0.367 (0.489) | 0.130 (0.070) | 18401.000 (16735.612) | 97.4% (7.7%) |
| GaussianCov | MaxCov | 0.000 (0.000) | 256.074 (301.431) | 0.098 (0.098) | 0.123 (0.053) | 12036.871 (7137.082) | 99.7% (0.9%) |
| | TOP | 1845.730 (1377.287) | 1004.588 (1728.804) | 0.344 (0.597) | 0.091 (0.042) | 14231.619 (14177.320) | 99.0% (2.9%) |
| | UniCov | 0.000 (0.000) | 237.247 (299.693) | 0.086 (0.098) | 0.116 (0.031) | 10157.665 (6907.177) | 100.0% (0.1%) |
| | Brownian | 0.627 (0.126) | 1029.781 (1240.544) | 0.372 (0.400) | 0.206 (0.130) | 24515.343 (15633.415) | 96.2% (9.6%) |

Table 8: Results with Gaussian sensor strategies (Table 2)

## G   LICENSING DATASETS & ASSETS USED

We use several publicly available datasets in this study and ensure that each is properly credited and used in accordance with its license and usage terms. For the Sim2Real-Fire dataset, we use simulation data, which is released under the Apache-2.0 license as noted in the original publication and hosted repository (available at `https://github.com/TJU-IDVLab/Sim2Real-Fire`). The Burn Probability (BP) for the United States (270-m), version 2023 (4th Edition) dataset, released by the USDA Forest Service, is publicly accessible for research and other uses under a Creative Commons Attribution (CC BY) license. It is available at `https://www.fs.usda.gov/rds/archive/`. We also incorporate the Fire Program Analysis Fire-Occurrence Database (FPA FOD), maintained by the U.S. Forest Service, which is available for public research under open data access policy (CC0 1.0 License). All cited datasets are accompanied by their corresponding original publications in our bibliography.

A summary of the asset licenses and their corresponding source web locations is provided in Table 9.

| Dataset | URL or DOI | License |
|---|---|---|
| National USFS Fire Occurrence Point (Feature Layer) | `https://data-usfs.hub.arcgis.com/datasets/6059c1a4dca749d393e33ee5f8a0cbaf_9/about` | CC0 1.0 |
| Sim2Real-Fire | `https://github.com/TJU-IDVLab/Sim2Real-Fire` | Apache-2.0 |
| Wildfire Hazard Potential for the United States (270-m), version 2023 (4th Edition) | `10.2737/RDS-2015-0047-4` | Creative Commons CC-BY |

Table 9: Overview of the used asset licenses and their corresponding source web locations.

## H   INSTRUCTIONS TO DOWNLOAD DATASET AND CODE

Our dataset and code are available in the supplementary material. Please refer to the README file for how to run the code and reproduce the results of our experiments.

## I   DATASET PLOTS

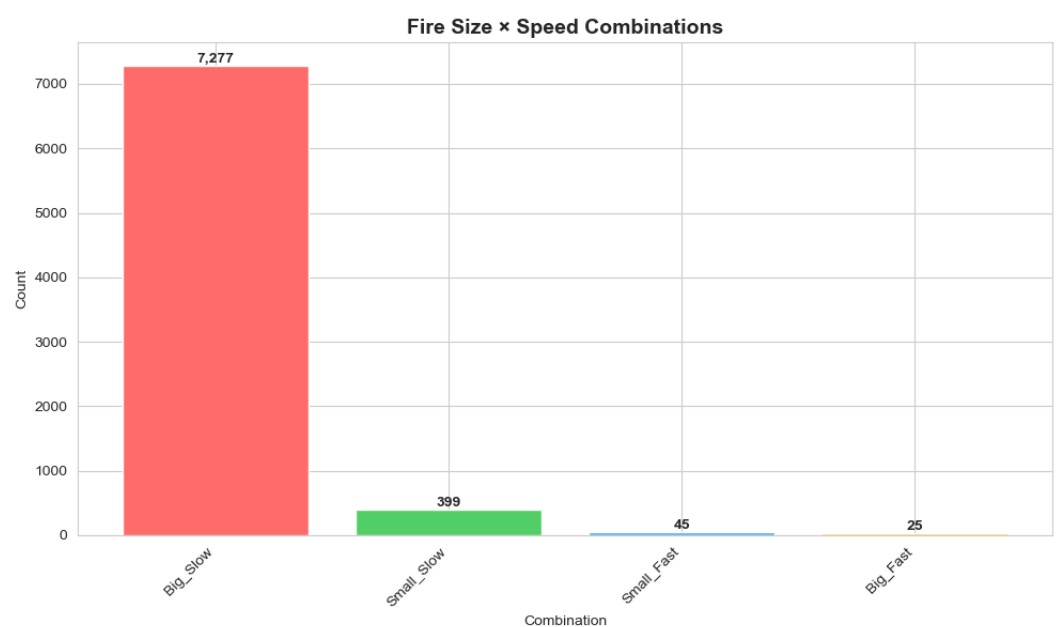

Figure 3: Distribution of fire sizes and speed in our dataset.

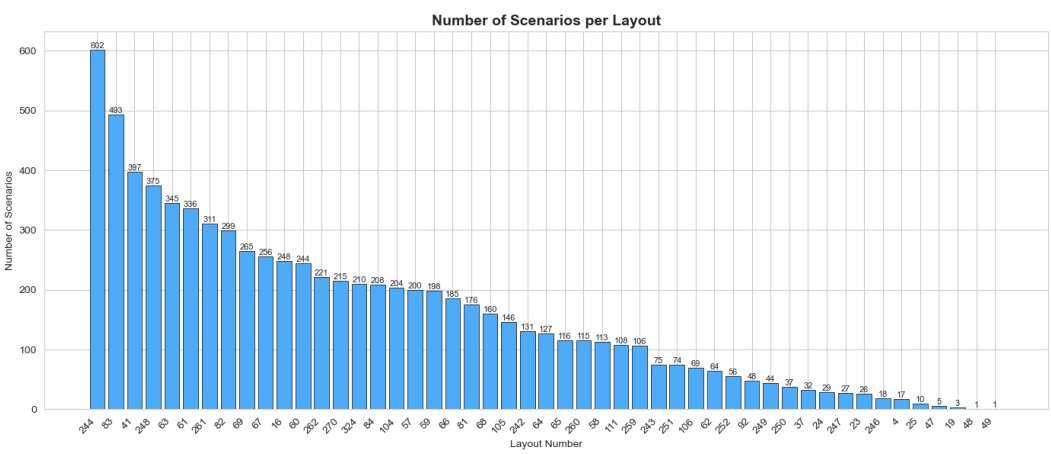

Figure 4: Number of wildfire scenarios per layout in our dataset.

# J ADDITIONAL VISUALIZATIONS

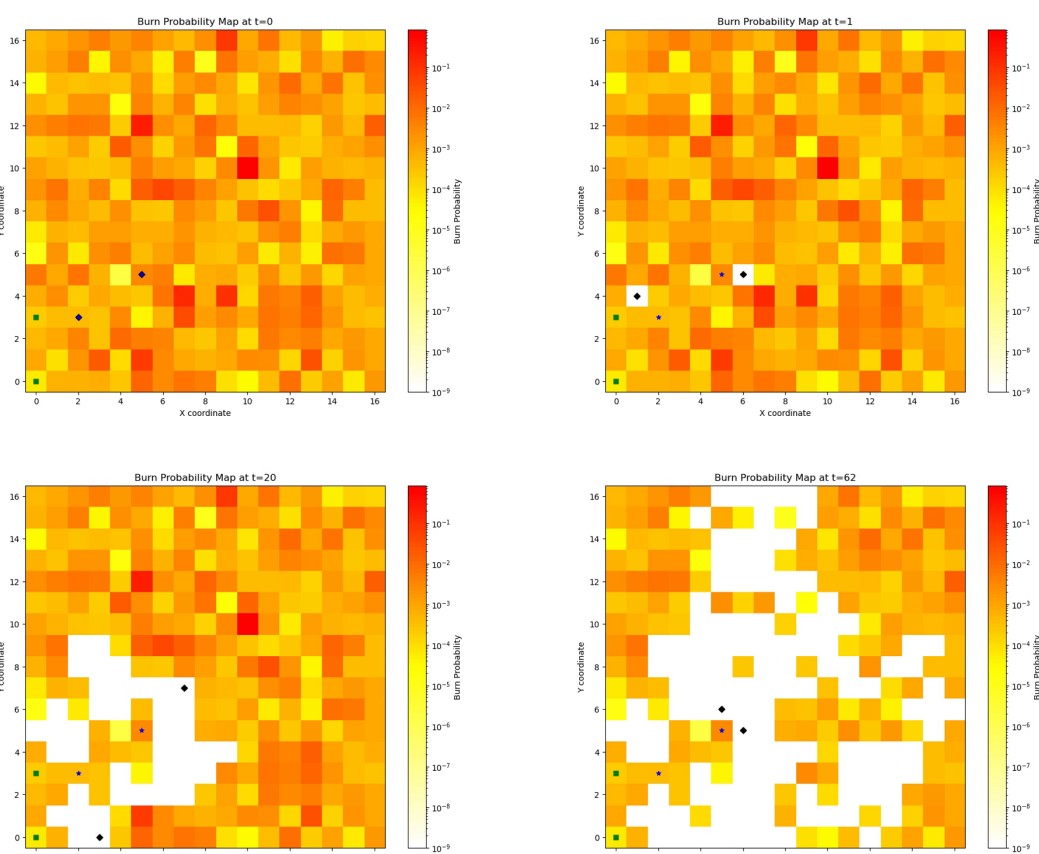

Figure 5: Visualization of ground sensors (green squares), charging stations (blue stars), and drone (black diamonds) trajectories overlaid on a small risk map. In this example, each of the two drones starts at a different charging station, but they both finish at the same one. The passage of the drones temporarily sets the visited cells' burn risk to 0. These images are frames extracted from a video generated using a function of our library.

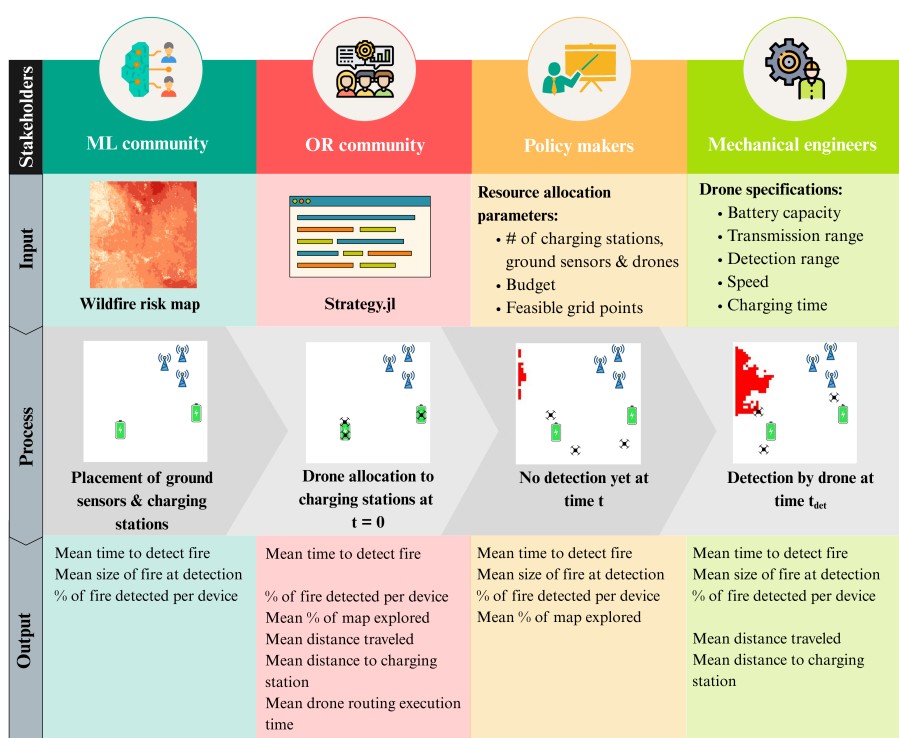

Figure 6: The `WFDroneBench` workflow supports four key stakeholder groups, each engaging with different system components. ML researchers benchmark AI-based wildfire risk maps via their effect on detection with fixed routing strategy; OR experts compare routing models using a fixed risk map; policy makers analyze resource allocation trade-offs; and ME engineers assess drone specification impacts. The figure outlines relevant inputs, decisions, and output metrics for each group.

