# OpenReview forum: "WFDroneBench: A Benchmark for Sensor Placement and Drone Routing for Wildfire Detection"
_ICLR.cc/2026/Conference — Submitted to ICLR 2026_

### Official Review · Reviewer_7yan · 2025-10-29

**Soundness:** 2
**Presentation:** 1
**Contribution:** 2
**Rating:** 2
**Confidence:** 5

**Summary:**

The authors have a good idea in trying to develop a risk-based approach to determining wildfire fighting strategies, but this paper is trying to do too much. It was very hard to follow.

**Strengths:**

The motivation is relevant and significant.

**Weaknesses:**

*The paper was hard to follow, the authors need to focus either on 1) validating whether their WFDroneBench workflow supports four key stakeholder groups as they claim, or 2) their benchmarking (including validating their semi-synthetic wildfire benchmark dataset.) For either case, they must get feedback from actual firefighters that their approach is actually effective.

**Questions:**

What research questions were the experiments answering?

How can firefighters know that your risk estimation is in alignment with actual operations?

---

> ### Author Response · Authors · 2025-11-17
>
> We acknowledge your comments and have put substantial time and care into refining the scope of the paper.
>
> We agree that the original framing attempted to cover too many perspectives. As Reviewer 2 also noted, the broad stakeholder positioning created confusion about the paper’s core contribution. In response, we have moved the stakeholder figure to the appendix and now focus the paper squarely on _benchmarking_ and on validating the semi-synthetic wildfire dataset. The additional quantitative analyses requested by Reviewer 3, such as distributions of fire types and data-quality metrics, are being added to strengthen this component.
>
> Regarding firefighter feedback, our work builds on established literature demonstrating the use of drones for wildfire detection, and our contribution is in providing a rigorous benchmark and evaluation framework rather than proposing novel firefighting operational concepts.
>
> We hope this new framing addresses your concern. Please let us know if some points still need changes. We are more than happy to engage in a constructive discussion with you integrate your feedback to make this paper suitable for you to recommend it.
> We will upload the updated version to OpenReview shortly.

---

### Official Review · Reviewer_MC1z · 2025-10-31

**Soundness:** 2
**Presentation:** 2
**Contribution:** 3
**Rating:** 4
**Confidence:** 3

**Summary:**

This paper proposes an open-source benchmark WFDroneBench for early fire detection, incorporating risk map, deployment strategies of ground equipment and drone routing strategies. WFDroneBench is modular and supports separate research on different system components. In addition, the paper curates a wildfire dataset containing 7746 scenarios. This paper evaluates and compares different wildfire detection strategies in its benchmark.

**Strengths:**

1. Good motivation. Proactive wildfire detection proves valuable in wildfire research.
2. The modular workflow allows the system’s components to support diverse research objectives, enhancing the extensibility of this benchmark.
3. The paper is well-written and provides a detailed introduction to the proposed benchmark.

**Weaknesses:**

1. The experiment results are inconsistent with the discussion conclusions:
* In Table 2, the UniCov drone routing strategy achieved the optimal detection rates and comparable detection times compared to TOP and Max-Cov, which contradicts the conclusion in line 413.
* In Table 2, compared with Max-Cov, UniCov achieved better detection rate and detection time simultaneously under the Rondom Sensor Strategy. The same experimental result is shown in sub-scenarios such as Slow Big and Fast Big, which contradicts the conclusion in line 418.
* In Table 2, compared with Uni-Cov, a significant decline in the detection rates of TOP and Max-Cov is observed, especially in Slow Small and Fast Big scenarios. The negative impact on detection rates may outweigh the benefits of the risk map.
* When changing from the GT dynamic map to the BP risk map, the performance of TOP has declined, especially in the Fast Big scenario. This reult cannot prove robustness proposed in line 431.
* In Table 3, Max-Cov outperforms Uniform Coverage in detection rate, which contradicts the conclusion in line 436.
2. With respect to the proposed dataset, additional quantitative analyses of data distribution and data quality are recommended.

**Questions:**

It would be valuable to vary the numbers of ground stations, charging stations, and drones to conduct comparative experiments for exploring optimal resource allocation.

---

> ### Author Response · Authors · 2025-11-17
>
> Thank you very much for your detailed and thoughtful review. We truly appreciate the time and care you put into it, and we have invested significant effort in addressing the points you raised.
>
> > The experiment results are inconsistent with the discussion conclusions
>
> Thank you for raising this. We identified two points we did not make explicit in our presentation of the results, making them inconsistent. We made these two points explicit in the updated version of the paper.
>
> (1) First, our discussions implicitly focused on the ground-truth map. The following passages that you pointed as contradictory actually hold for the 'ground-truth' maps (Table 3) but not for the 'BP' ones (Table 2):
> > In Table 2, UniCov achieves optimal detection rates and comparable detection times, contradicting line 413.
>
>  >In Table 2, UniCov outperforms Max-Cov under Random Sensors, contradicting line 418.
>
>  We now make it explicit that these conclusions hold for the ground-truth map.
>  Concerning the following:
> > TOP and Max-Cov experience significant declines compared to UniCov, raising concerns about whether risk maps help.
>
> This decline is expected when switching from an oracle map to imperfect risk maps. We now clarify that our conclusions about the benefit of risk maps apply only when risk maps are accurate, and we provide the distribution of fire sizes and quality metrics in the appendix to contextualize these declines. UniCov serves as a sanity-check baseline when risk maps are weak.
>
> (2) Second, we implicitly used detection-time as 'performance' (when claiming that one strategy 'outperforms' or is 'robust') while you rightfully considered both detection time *and* detection percentage as a measure of 'performance'. This applied for your following remarks:
> > TOP declines when moving from GT to BP maps, especially in Fast Big; this contradicts robustness claimed in line 431.
>
> > In Table 3, Max-Cov outperforms Uniform Coverage, contradicting line 436
>
> We now precise that we mean robustness in terms of detection time, and state clearly that MaxCoverage generally offers faster detection, while Uniform Coverage may achieve competitive or superior detection rates *depending on risk-map quality*.
>
> In addition to your comment about the results discussion, you also offered the following feedback:
> > With respect to the proposed dataset, additional quantitative analyses of data distribution and data quality are recommended.
>
> Thank you for this suggestion. We agree that additional quantitative analysis of the data distribution and data quality will strengthen the dataset component of the paper. We are currently preparing new histograms showing the distribution of small versus big fires, along with counts by fire category and relevant quality metrics. These analyses will be added to the appendix, and we will follow up with the updated figures as soon as they are finalized.
>
> Finally, you asked the question:
> > It would be valuable to vary the numbers of ground stations, charging stations, and drones to conduct comparative experiments for exploring optimal resource allocation.
>
> Thank you for raising this point. We agree that exploring different numbers of ground stations and drones is valuable, and we have experimented with a few such combinations during development. However, a comprehensive resource-allocation analysis would significantly expand the scope of the paper, and we believe it is better positioned as follow-up work building on this benchmark. We are currently pursuing this direction in a follow-up work and welcome your thoughts on which resource dimensions you consider most impactful.
>
>
> Thank you again for the time you spent reviewing the paper. We believe your points (especially about the results discussion) helped significantly improving the clarity of the paper.
> We hope our modifications address your concerns. If so, we would appreciate it if you could reflect it in your assessment. Please let us know if some points still need clarifications or changes. We are happy to continue the discussion and update the paper if needed.
> We will upload the updated version to OpenReview shortly.

---

> ### Author Response · Authors · 2025-11-27
>
> Dear Reviewer,
>
> Your comments were helpful to improve our paper. We uploaded a revised version integrating the changes. We would be grateful to hear your thoughts on our answer and this new version.
> We are available should you need any further information.
>
> Best Regards,
> The Authors

---

### Official Review · Reviewer_8HFm · 2025-10-31

**Soundness:** 3
**Presentation:** 2
**Contribution:** 2
**Rating:** 6
**Confidence:** 4

**Summary:**

This work proposes a new benchmarking framework, WFDroneBench, for early wildfire detction. It integrates a wide range of scenarios and locations based on historical ignition data and physics-based spread simulations. Hereby, the authors aim to provide a benchmark that supports the interests of multiple stakeholders and viewpoints, covering, e.g., the detection, drone allocation, and mechanical engineering decisions. In their experiments, they show the performance of different strategies in sensor placement and drone routing in wildfire detection.

**Strengths:**

- The paper is written in a very clear and concise way. It is easy to follow the proposed ideas and setup of the benchmark.
- WFDroneBench allows evaluating multiple important factors of the wildfire detection tasks, such as charging station and ground sensor placements, on top of the drone operations themselves. Hence, a holistic evaluation of strategies can be obtained.
- The benchmark is equipped with a selection of routing and sensor placement baselines to compare with.

**Weaknesses:**

- While the evaluations consider different strategies, only a single hardware setup is demonstrated. Showing additional drone hardware configurations would support the claims about meeting the needs of, e.g., mechanical engineers, as claimed in Figure 1.
- While the included strategies for drone routing and sensor placement act as a baseline (e.g., uniform coverage, brownian motion), the experiments section would be enhanced significantly by incorporating other, state-of-the-art drone routing approaches.
- The evaluation of the benchmark focuses on the rate of fires detected in under 12 hours (Table 2, 3) with varying sensor placement and routing. It seems like the proposed utility of the benchmark for policy makers and mechanical engineers seems underexplored by the manuscript.

**Questions:**

- In line 471, it is mentioned that environmental factors such as wind and temperature are not considered in the evaluation of the drones' performance. Since these factors impact the spread of wildfires, and may be useful for tasks such as routing, do the scenarios in the benchmark contain these informations, or is their inclusion entirely up to future work?

---

> ### Author Response · Authors · 2025-11-17
>
> We sincerely appreciate your review and have put substantial time and care into addressing all suggested changes. Thank you for highlighting the scope of our paper, which helped us significantly sharpen our focus on the core contributions.
>
> > While the evaluations consider different strategies, only a single hardware setup is demonstrated. Showing additional drone hardware configurations would support the claims about meeting the needs of, e.g., mechanical engineers, as claimed in Figure 1.
>
> Thank you for pointing this out. We agree that Figure 1 may overstate the breadth of applicability to mechanical engineers, policymakers, and other stakeholders. To avoid overstating the scope, and to keep the paper focused rather than adding many additional experiments for multiple communities, we narrow the framing of the figure and move it to the appendix. The main paper now focuses more clearly on the technical and scientific contributions to the ML community, and insights from our experiments.
>
> > The evaluation of the benchmark focuses on the rate of fires detected in under 12 hours (Table 2, 3) with varying sensor placement and routing. It seems like the proposed utility of the benchmark for policy makers and mechanical engineers seems underexplored by the manuscript.
>
> Thank you again for noting this. As stated above, we refined the scope of our paper to reduce the emphasis on policy makers and mechanical engineers and focus on the ML community. Nonetheless, we have strengthened the results section by explicitly highlighting broadly applicable insights revealed by the evaluations, such as the relative value of drones versus sensor deployments, the dependence on risk-map accuracy, and the trade-off between detection speed and coverage.
>
> > While the included strategies for drone routing and sensor placement act as a baseline (e.g., uniform coverage, brownian motion), the experiments section would be enhanced significantly by incorporating other, state-of-the-art drone routing approaches.
>
> We appreciate the request for other advanced routing methods. To clarify, in our problem setting, the two state-of-the-art algorithmic families applicable at scale are TOP-based routing and MaxCoverage; these form the backbone of current operational and research-grade approaches for persistent aerial monitoring. We decided to keep a RL approach for future work. The remaining strategies we include, such as uniform coverage and Brownian motion, serve as diagnostic baselines for evaluating burn-map quality and highlighting open-problem behavior.
>
> Regarding your question:
>
> > Since [weather and environmental] factors impact the spread of wildfires, and may be useful for tasks such as routing, do the scenarios in the benchmark contain these informations, or is their inclusion entirely up to future work?
>
> Yes, our dataset does contain wind, temperature and weather information for each scenario, in the associated `Weather_Data` folder.
>
> Thank you again for your careful review that helped us improve the paper. We hope that our modifications and clarifications address the weaknesses you identified. If you believe that our modifications convincingly addressed your concerns, we would appreciate if you could consider reflecting it in your assessment. Please let us know if some points still need clarifications or changes. We are happy to continue the discussion and update the paper if needed.
> We will upload the updated version to OpenReview shortly.

---

> > ### Author Response · Authors · 2025-11-27
> >
> > Dear Reviewer,
> >
> > Your comments were helpful to improve our paper, and we would be grateful to hear your thoughts on our rebuttal. We are available should you need any further information.
> >
> > Best Regards,
> > The Authors

---

### Official Review · Reviewer_6G5W · 2025-10-31

**Soundness:** 3
**Presentation:** 2
**Contribution:** 2
**Rating:** 4
**Confidence:** 2

**Summary:**

This paper introduces WFDroneBench, an open-source benchmark and simulation framework that helps researchers and engineers test and compare wildfire detection strategies. The benchmark is extensively and useful, using real ignition data and realistic fire spread simulations. Experiments show that risk-aware routing strategies — especially TOP and Max-Coverage — detect fires faster and more effectively than random or uniform approaches. Drones outperform stationary sensors by a large margin. However, the success of these strategies depends heavily on having accurate and up-to-date risk maps.

**Strengths:**

The benchmark appears to be well executed and comprehensive.

1. Wildfires are a critical global issue, and early detection is both societally impactful and technically challenging. Thus a benchmark for this task is likely going to be very useful with the rise in global warming.
2. Glad to see that the benchmark would be completely open-source and has an associated library and toolkit, hopefully would be used to explore this clearly large problem space.
3. The experiments evaluate multiple baselines under consistent conditions, including both static and dynamic risk maps. The authors appear to have put effort in constructing reasonable baselines.

**Weaknesses:**

1. The work is primarily a benchmark, not a new algorithmic contribution. The routing formulations are not conceptually new. Reinforcement learning or imitation learning methods are only mentioned as future work. Including even one such baseline would make the benchmark more forward-looking.
2. The text occasionally reads more like a system report — it’s dense, with many details about parameters and datasets but less emphasis on insights.
3. Relevance to ICLR is low. It fits better in a systems, applied AI, or robotics venue than a core learning venue. Please comment on why this work is relevant for this venue.

**Questions:**

Please clarify my questions in weakness. In general, the paper is well executed but doesn't offer many insights into the problem itself and could use some experiments which illuminate open problems in this domain which would improve the benchmark.

---

> ### Author Response · Authors · 2025-11-17
>
> Thank you very much for your careful review. We devoted significant time and effort to address the points raised.
>
> >The work is primarily a benchmark, not a new algorithmic contribution.
>
>  While our work is presented as a benchmark, it also includes several non-trivial algorithmic contributions. Specifically, we performed several extensions to the Team Orienteering Problem (TOP) as the classical setting did not apply to our problem. Our modified formulation supports multiple starting and ending depots, with constraints on the number of drones at each depot, and allows revisiting the same location. We also modified the Particle Swarm Optimization (PSO) algorithm to adapt it to grid instances, achieving up to 400× speedup.
>
> > The text \[...] is dense, with many details about parameters and datasets but less emphasis on insights.
>
>  While we have already emphasized several insights in the results section, we agree that the text can better foreground the scientific takeaways. We now highlight key insights more explicitly: drones are dramatically more effective than fixed sensors for early detection (one drone can survey ~60 cells per hour, whereas a stationary sensor covers only its own cell), contradicting current real-world reliance on ground sensors. Our results also clarify the trade-off between detection rate and detection speed across strategies, and emphasize the critical role of risk map quality. We revised the text to make these insights more prominent and easier to digest.
>
> > Please comment on why this work is relevant for this venue.
>
> Thank you for giving us the opportunity to argue for the fit of our paper. The ICLR'2026 call for papers explicitly list `datasets and benchmarks` as an item in its list of relevant topics, as well as `applications to planning`. We position our work as a dataset and benchmark paper aimed specifically at the machine learning community similar to some work recently published at ICLR like [1, 2, 3, 4]
> We believe the problem setting we present opens several impactful ML research directions. In particular, our results reveal two major open challenges: (i) detecting small fires rapidly and reliably, and (ii) improving risk-map prediction, where the gap between ground-truth ignition patterns and available risk maps highlights a significant opportunity for ML innovation.
> Wildfire early detection is a pressing and globally relevant problem with increasing interest across ML, applied AI, and climate-related modeling, and we believe that providing a principled, extensible benchmark is a valuable contribution for the ICLR community.
>
> We hope that our replies clarify the key insights and open problems, and we hope they address the weaknesses you identified. Please let us know if some points still need clarifications or changes. If you believe we convincingly addressed your concerns, please consider reflecting it in your assessment. We are happy to continue the discussion and update the paper if needed.
> We will upload the updated version to OpenReview shortly.
>
> References:
>
> [1] Tec, Mauricio, et al. "SpaCE: The Spatial Confounding Environment." ICLR'2024.
> [2] Fortier, Matthew, et al. "CarbonSense: A Multimodal Dataset and Baseline for Carbon Flux Modelling." ICLR'2025.
> [3] Wang, Xiangyu, et al. "Towards Realistic UAV Vision-Language Navigation: Platform, Benchmark, and Methodology." ICLR'2025.
> [4] Stein, Gideon, et al. "CausalRivers-Scaling up benchmarking of causal discovery for real-world time-series." ICLR'2025.

---

> > ### Comment · Reviewer_6G5W · 2025-11-17
> > **Learning baselines**
> >
> > I appreciate the response, I think I’m reasonably convinced by the response. I noticed that the paper doesn’t include learning based baselines — either using IL or RL. It would be very useful for the community if such a baseline (and associated code) is included, along with depicting some of their failure cases, as that would lead other researchers from actually implementing algorithmic improvements. Any response to that concern?

---

> > > ### Author Response · Authors · 2025-11-17
> > >
> > > We truly appreciate your quick response!
> > > We completely agree that RL/IL baselines would be valuable additions to the benchmark. However, implementing a well-performing learning-based approach requires substantial effort: designing appropriate state/action representations for the spatiotemporal fire detection problem, extensive hyperparameter tuning, and sufficient training to ensure the baseline is competitive rather than strawman. Given the tight rebuttal timeline, we are concerned that a hastily implemented learning baseline might not do justice to these methods and could even misrepresent their potential.
> > > That said, we want to be responsive to the community's needs. We see two paths forward:
> > >
> > > (1) Include in future work: We explicitly frame RL/IL baselines as a key direction for future research in the paper. This positions the benchmark as an open platform for the community to contribute learning-based methods.
> > > (2) Attempt implementation: If you believe an RL/IL baseline is essential for acceptance, we are willing to attempt a basic implementation within the remaining time, with the understanding that it may be preliminary and require further refinement.
> > >
> > > Could you please advise which approach you think would be most appropriate? We want to ensure our response aligns with your expectations while maintaining the quality standards of the benchmark.
> > > We appreciate your continued engagement and constructive feedback.

---

> > > > ### Author Response · Authors · 2025-11-27
> > > >
> > > > Dear Reviewer,
> > > >
> > > > As the discussion period is coming to an end, we wanted to follow up on our previous message to ensure we have enough time to update the paper should any further revisions be needed.
> > > >
> > > > Best regards,
> > > > The Authors

---

### Meta-Review · Area_Chair_NJx7 · 2026-01-11

**Summary:**

This paper introduces WFDroneBench, an open-source benchmarking framework for optimizing sensor placement and drone routing in early wildfire detection. It integrates machine-learned risk maps with optimization strategies, evaluates various detection approaches across thousands of scenarios, and highlights the superiority of risk-aware routing methods when accurate risk maps are available.

Reviewers raise some key concerns:

1. Reviewer 6G5W expressed that the work is primarily a benchmark with limited algorithmic novelty, and noted the absence of learning-based baselines (e.g., RL/IL). He/She also questioned the paper’s relevance to ICLR, suggesting it may be more suited to applied AI or robotics venues.

2. Reviewer 8HFm highlighted that evaluations used only a single hardware setup, limiting claims of broad applicability. He/She also suggested including more state-of-the-art routing methods and felt the utility for policymakers and engineers was underexplored.

3. Reviewer MC1z pointed out apparent inconsistencies between experimental results and stated conclusions, particularly regarding performance comparisons between strategies. They also recommended more quantitative analysis of the dataset distribution and quality.

4. Reviewer 7yan found the paper unfocused and hard to follow, suggesting it should either validate the benchmark’s support for claimed stakeholders or rigorously evaluate the dataset—preferably with feedback from actual firefighters to ensure practical relevance.

These concerns collectively informed the suggested decision, emphasizing the need for clearer framing, stronger experimental validation, and better alignment with the conference’s focus on machine learning contributions. According to this feedback, I cannot recommend the acceptance for this version.

**Reviewer Concerns:**

Based on the authors' rebuttal tried to address the following reviewer concerns:

(1) Reviewer 6G5W’s relevance concern: The authors clarified that ICLR explicitly includes datasets/benchmarks and applications to planning, citing recent ICLR papers as precedent. They also sharpened the focus on ML research gaps (e.g., risk-map prediction, small-fire detection).

(2) Reviewer 6G5W’s request for more insights: The authors updated the text to explicitly highlight key scientific takeaways, such as drones being dramatically more effective than fixed sensors and the critical role of risk-map quality.

(3) Reviewer 8HFm’s overstatement of scope: The authors narrowed the framing, moved the broad stakeholder figure to the appendix, and refocused the paper on ML contributions rather than mechanical engineering or policy.

(4) Reviewer 8HFm’s question about environmental data: The authors confirmed that wind, temperature, and weather data are included in the dataset.

(5) Reviewer MC1z’s inconsistency in results interpretation: The authors clarified that certain conclusions were specific to ground-truth maps (not imperfect risk maps) and refined the language regarding performance metrics (detection time vs. detection rate).

(6) Reviewer 7yan’s lack of focus: The authors streamlined the paper’s scope to focus on benchmarking and dataset validation, removing overambitious stakeholder claims.

However, some concerns are not well addressed:

(1) Reviewer 6G5W’s request for learning-based baselines (RL/IL): The authors acknowledged their value but deferred implementation to future work due to time constraints, offering only a tentative commitment to add a basic version if required for acceptance.

(2) Reviewer 8HFm’s request for additional hardware configurations and SOTA routing methods: Not directly addressed; the authors stated that TOP and MaxCoverage represent current SOTA for this problem and kept other baselines as diagnostic.

(3) Reviewer MC1z’s request for quantitative dataset analysis: The authors promised to add distribution histograms and quality metrics in the appendix, but these were not shown in the rebuttal.

(4) Reviewer MC1z’s suggestion to vary resource counts (drones, stations): Deferred to follow-up work.

(5) Reviewer 7yan’s request for firefighter feedback or operational validation: Not addressed beyond stating the work builds on existing literature; no plans for real-world validation were mentioned.

**Reviewer Scores:**

Reviewer 6G5W initially scored the paper marginally below acceptance. Their engagement in the rebuttal suggested they were largely convinced by the authors’ clarifications on relevance and insights. While they noted the lack of learning-based baselines, their constructive tone indicates they likely would have raised their score to a 5 after a full discussion.

Reviewer 8HFm gave a marginally positive score. Their main concerns about scope and environmental data were replied in the rebuttal. However, the concerns about additional drone hardware configurations are not addressed. The reviewer may keep the initial score.

Reviewer MC1z scored the paper marginally below acceptance due to perceived inconsistencies in results presentation. The authors’ point-by-point clarifications and promised dataset analyses directly resolved these issues. Given the constructive tone, they may increased their score.

Reviewer 7yan firmly rejected the paper for being unfocused and lacking real-world validation. The authors narrowed the scope but did not address the need for firefighter feedback or operational proof. Therefore, this reviewer’s score would likely have remained at a 2, or seen only a minimal increase to a 3.

---

### Decision · Program_Chairs · 2026-01-26

Reject